# Effect of Team-Based Learning Interventions on the Learning Competency of Korean Nursing Students: A Systematic Review and Meta-Analysis

**DOI:** 10.3390/bs15030390

**Published:** 2025-03-19

**Authors:** Kawoun Seo, Seyeon Park

**Affiliations:** 1Department of Nursing, Joongbu University, Geumsan 32713, Republic of Korea; kwseo@joongbu.ac.kr; 2College of Nursing, Chungnam National University, Daejeon 35015, Republic of Korea

**Keywords:** students, nursing, team-based learning, learning competency, systematic review, meta-analysis

## Abstract

This systematic review and meta-analysis aimed to comprehensively evaluate the effectiveness of team-based learning (TBL) methods in Korean nursing education, focusing on their impact on students’ key learning competencies. Specifically, this study examined the effects of TBL on communication ability, self-efficacy, learning ability, and problem-solving skills while also assessing the overall efficacy of TBL compared to other teaching methods. This was a systematic review and meta-analysis of experimental and quasi-experimental studies. The relevant literature was sourced from Medline/PubMed, Embase, Cochrane Library, Cumulative Index to Nursing and Allied Health Literature Complete, Google Scholar, and the reference lists of the retrieved papers. The literature was selected based on the Preferred Reporting Items for Systematic Reviews and Meta-analysis guidelines. The risk of bias was assessed using the Cochrane risk of bias assessment tool and the Risk of Bias for Nonrandomized Studies tool. Standardized mean differences and a random effects model were employed to determine effect sizes. A total of twelve studies met the inclusion criteria and were analyzed. The overall standardized mean difference was 0.69 (confidence interval = 0.43–0.94, *p* < 0.001), indicating a moderate to strong effect of TBL on nursing students’ learning competencies. However, due to a substantial degree of heterogeneity (89%), subgroup analyses were conducted to assess the impact of TBL on specific learning competencies. The standardized mean difference for communication ability, self-efficacy, and learning ability was 0.74 (confidence interval = 0.22–1.26, *p* = 0.004), 0.32 (confidence interval = 0.13–0.51, *p* = 0.005), and 0.44 (confidence interval = 0.26–0.63, *p* < 0.001), respectively. Notably, TBL had the most significant impact on problem-solving ability, with a standardized mean difference of 1.10 (confidence interval = 0.37–1.83, *p* = 0.006), suggesting that TBL is particularly effective in fostering problem-solving skills among nursing students. All the findings were statistically significant. Team-based learning interventions have a substantial positive impact on key learning competencies in nursing education, particularly in enhancing problem-solving skills. However, while TBL demonstrates clear benefits, current evidence remains insufficient to definitively establish its superiority over alternative instructional methods. Further research is needed to compare TBL with other active learning strategies and to explore its long-term effectiveness in clinical and academic settings.

## 1. Introduction

In the realm of education, there has been a rise in the adoption of innovative teaching approaches that diverge from conventional teacher-centric models and foster student-centered autonomous learning ([8]). Methodologies such as problem-based learning, blended learning, flipped learning, and team-based learning (TBL) are being emphasized to enhance students’ critical thinking, problem-solving, and communication skills. These methodologies incorporate collaborative processes and group activities to bolster individual competency. TBL, in particular, stands out for its focus on team-based problem-solving over individual efforts.

Numerous studies have demonstrated that TBL significantly enhances team dynamics by fostering collaborative learning, communication, and leadership skills among nursing students ([1]; [4]; [14]). The structured nature of TBL promotes active participation, peer learning, and accountability, which are crucial for effective teamwork in clinical settings ([4]). Additionally, TBL has been shown to improve problem-solving and critical thinking skills by engaging students in case-based discussions and decision-making tasks ([7]; [14]). These cognitive skills are particularly essential in nursing education, where students must develop the ability to assess patient conditions, prioritize care, and respond to complex clinical scenarios.

Despite the growing adoption of TBL in nursing education, there is limited synthesized evidence regarding its overall effectiveness and impact on learning outcomes. Therefore, this study aims to systematically review and analyze intervention studies that implemented TBL in nursing education to determine its efficacy. Specifically, we seek to address the following research questions: (1) What are the measurable effects of TBL on nursing students’ cognitive, affective, and skill-based competencies? (2) How does TBL compare to other instructional methods in enhancing nursing education? (3) What factors contribute to the variability in TBL’s effectiveness across different studies?

We systematically reviewed intervention studies that implemented TBL teaching methods based on the Participants, Intervention, Comparison, and Outcome (PICO) model. Additionally, we conducted a meta-analysis to ensure the quantitative adequacy of the studies reviewed, confirmed heterogeneity, and performed subgroup analyses.

## 2. Methods

This systematic review and meta-analysis were conducted according to the Preferred Reporting Items for Systematic Reviews and Meta-Analyses (PRISMA) 2020 statement (Appendix A) ([11]).

### 2.1. Search Strategy

The literature search was conducted over three months, including preliminary searches and strategy refinement, with the final systematic search performed from 1 October to 3 October 2023. Following Cochrane Library guidelines, a short final search period (1–3 days) was used to ensure study reproducibility while capturing the most recent publications.

We searched Medline/PubMed, Embase, Cochrane Library, CINAHL Complete, and Google Scholar using both electronic and manual methods. Search terms included variations in “team-based learning” and “nursing education”.

The search period (January 2013–June 2023) was chosen to ensure relevance to recent teaching methodologies and technological advancements in TBL. Reference lists and gray literature were also reviewed to minimize publication bias.

### 2.2. Selection of Studies

The primary research inquiries were conducted based on Participants, Intervention, Comparison, Outcome, Time, Setting, and Study Design (PICOTS-SD) criteria, as shown below:Population: Nursing undergraduate students;Intervention: Implementation of a TBL program;Comparison: Previous employment of traditional educational methods, unrestricted;Outcomes: Enhancement of learning competencies, particularly communication, self-efficacy, learning aptitude, problem-solving skills, and critical thinking;Timing: Pre- and post-intervention assessment;Setting: No specific constraints;Study Design: Randomized and non-randomized controlled trial experimental designs.

### 2.3. Inclusion and Exclusion Criteria

The literature was selected based on the PRISMA guidelines. The inclusion criteria encompassed academic journals, dissertations, and randomized and non-randomized controlled experimental studies that investigated team-based learning (TBL) educational interventions aimed at enhancing the learning competency of undergraduate nursing students. To ensure a comprehensive analysis and reduce publication bias, we included studies published in both English and Korean, as the majority of TBL research in nursing education during the targeted period was conducted in Korea. This clarification has now been explicitly stated in the Methods section.

### 2.4. Assessment of the Risk of Bias

Two researchers independently assessed the risk of bias in the selected studies. Disagreements were resolved through discussions, with agreed-upon outcomes presented in the study. Randomized controlled trials (RCTs) were evaluated based on the six domains outlined in the Cochrane Risk of Bias tool: randomization process, deviations from the intended interventions, measurement of the outcome, missing outcome data, selection of the reported results, and overall bias. Non-RCTs were assessed based on the six domains of the Risk of Bias for Nonrandomized Studies tool: selection of participants, confounding variables, measurement of exposure, blinding of outcome assessment, incomplete outcome data, and selective outcome reporting. The risk of bias was categorized as unclear, low, or high.

### 2.5. Data Analysis

Two individuals, including the author of this study, independently extracted and coded data from the selected studies. Key details, such as the research topic, authorship, publication year, study design, grading, interventions, comparative measures, intervention duration, and outcome assessment tools, were meticulously cataloged using Microsoft Excel. Additionally, means, standard deviations, and participant counts before and after the interventions for each group were coded to facilitate calculations of effect size. The outcome variables were categorized as communication ability, self-efficacy, learning ability, or problem-solving ability. The coded data were analyzed using RevMan (Review Manager) version 5.4.1 software (Cochrane Collaboration, Oxford, UK), and standardized mean differences (SMDs) were calculated using Hedges’ g, which corrects for small sample size bias to provide more accurate effect size estimation. A random-effects model (REM) was applied to account for variability across studies. According to the Cochrane Handbook for Systematic Reviews of Interventions ([6]), REM is appropriate when substantial heterogeneity exists among studies due to differences in study populations, interventions, and outcome measures. Given the diverse TBL implementations in the included studies, REM was deemed suitable as it allows for variations in true effect sizes. Additionally, since the intervention effects were expected to differ based on student demographics, instructional approaches, and intervention duration, the model was chosen to accommodate this variability. The heterogeneity measure I^2^ was found to be 89%, indicating substantial heterogeneity, which further supports the use of REM.

Effect sizes were interpreted based on Cohen’s standards, with values of 0.2, 0.5, and 0.8 denoting small, medium, and large effects, respectively ([6]). Heterogeneity was assessed using both Q and I^2^ values, and two researchers independently reviewed the forest plot distribution to ensure the reliability of the analysis. To further explore sources of heterogeneity, subgroup analyses were conducted based on outcome variables and intervention characteristics, allowing for a more nuanced interpretation of the results.

To address potential outcome dependence, we carefully reviewed included studies to determine whether multiple effect sizes from the same study participants influenced the meta-analysis. Following [3] ([3]), if results from the same researcher or dataset examined different learning competencies as independent outcomes, they were considered separate effect sizes. Additionally, we ensured that the studies adhered to the PICO criteria for systematic review selection, reinforcing their validity. While outcome dependence is a recognized issue in meta-analyses, we maintain that the reliability of the standardized mean difference (SMD) is not compromised, as the statistical approach used accounts for variance within and across studies.

## 3. Results

### 3.1. Selection of Studies

Figure 1 illustrates the process of selecting the relevant literature. A total of 500 articles were obtained from electronic databases, and three articles were derived from manual searches. We organized this literature using bibliographic management software (EndNote 20, Clarivate™, Philadelphia, PA, USA). After removing 263 duplicate documents, we independently evaluated 240 articles by scrutinizing their titles and abstracts based on PICO criteria. There were differences in the number of studies selected. We reached a consensus through discussions and decided to select full-text papers. Subsequently, 208 documents were excluded, and 32 full-text papers were thoroughly examined. Based on inclusion and exclusion criteria, 20 articles were deemed ineligible, resulting in the selection of 12 articles. Finally, from the selected articles, we gleaned findings regarding the impact of TBL interventions on nursing students’ learning competency.

### 3.2. Characteristics of Studies

The literature review provided an overview of the literature by detailing key characteristics, such as authorship, publication year, country of origin, research methodology, subject matter, intervention type, duration, method, comparative analysis, and outcome assessment. The reviewed publications, spanning from 2015 to 2020, originated exclusively in Korea and comprised eight articles in Korean and four in English. The number of participants ranged from 63 to 183 across the studies. One study had a randomized control group experimental design, while the rest had a non-randomized control experimental design. Various types of TBL approaches were employed as the intervention, such as traditional TBL, TBL combined with simulation, problem-centered TBL, reflection, and legal aspect integration. The intervention duration ranged from one to six hours per session, occurring 1 to 13 times a week, and spanned from one week to one semester. Regarding comparative interventions, eight studies employed traditional, lecture-based educational interventions, while the rest implemented simulation-based lectures. Thirteen evaluation tools, including the Global Interpersonal Communication Competence Scale (GICC-15), were used for outcome assessment (Table 1).

### 3.3. Risk of Bias in Studies

We evaluated the risk of bias in the 12 selected studies. The study with a RCT design had a low risk of bias in the selection of research participants, selective result reporting, and incomplete outcome data. However, there was a risk of bias in concealing the allocation of experimental and control groups, and information on performance bias was not reported (Figure 2a,b).

The risk of bias assessment of the eleven studies employing a non-RCT design revealed that the risk of bias was low in the selection of participants. Given the non-randomized nature of the design, the selection of the control and experimental groups fell under the purview of the researchers. Regarding confounding variables and measurement bias, 75% of the studies exhibited a low risk, indicating a generally high quality of research. However, regarding the outcome evaluation bias, 40% of the studies demonstrated a low degree of bias, 40% exhibited a high degree of bias, and the remaining 20% showed an uncertain risk of bias. This underscores the varied ways in which intervention studies can be biased in terms of evaluating results. Notably, selective outcome reporting bias was low across all studies, reflecting excellent quality (Figure 3a,b).

### 3.4. Meta-Analysis

#### 3.4.1. Overall Effect Size

The overall effect size of TBL interventions on the learning competency of nursing students was medium or higher and statistically significant (SMD = 0.69, CI = 0.43–0.94, *p* < 0.001). However, as heterogeneity was found to be high (I^2^ = 89%, *p* < 0.001), the effect size was determined for specific learning competencies (Figure 4).

##### Effect Size for Communication Ability

The effect size for communication ability was SMD = 0.74 (CI = 0.22–1.26, *p* = 0.004). Six papers were included in this analysis, and heterogeneity was high (I^2^ = 90%, *p* < 0.005). This result stemmed from the diverse tools used to measure communication skills.

##### Effect Size for Self-Efficacy

The effect size for self-efficacy was statistically significant but the smallest among all learning competencies (SMD = 0.32, CI = 0.13–0.51, *p* = 0.005). Five studies were included in this analysis, and heterogeneity was found to be low (I^2^ = 16%, *p* = 0.005).

##### Effect Size for Learning Ability

The effect size for learning ability was medium and statistically significant (SMD = 0.44, CI = 0.26–0.63, *p* < 0.001). Four studies were included in the analysis, and heterogeneity was I^2^ = 0%, indicating that the studies reporting the effects on learning ability were almost homogeneous.

##### Effect Size for Problem-Solving Ability

Six studies were included in this analysis. The effect size for problem-solving ability was statistically significant and the highest among all learning competencies (SMD = 1.10, CI = 0.37–1.83, *p*-value = 0.006). However, a notable degree of heterogeneity was observed (I^2^ = 95%, *p* < 0.001). Examining the impact of TBL interventions on different learning capacities revealed compelling evidence.

#### 3.4.2. Subgroup Analysis

##### Frequency of Intervention

A subgroup analysis based on the frequency of intervention revealed that the effect of TBL interventions differs based on their frequency. Regarding communication ability, administering the intervention four or more times yielded a notably large but statistically insignificant effect, whereas administering it less than four times showed a smaller yet still effective improvement (Figure 5). The effect on self-efficacy was significantly larger when the intervention was administered less than four times (versus four or more times). The effect on problem-solving ability was significant in both groups, but the effect was larger when the intervention was administered less than four times. Conversely, the effect on learning ability was larger when the intervention was administered four or more times.

##### Type of Intervention

A subgroup analysis was conducted based on whether a traditional TBL intervention or a multicomponent TBL intervention (such as TBL combined with simulation or reflective methods) was administered (Figure 6). The effects on communication, learning, and problem-solving abilities were more pronounced when a traditional TBL intervention was administered. Interestingly, the effect on self-efficacy remained consistent across both types of TBL interventions.

## 4. Discussion

### 4.1. Summary of Key Findings

This systematic review and meta-analysis aimed to evaluate the impact of team-based learning (TBL) interventions on the key learning competencies of nursing students, including communication ability, self-efficacy, learning ability, and problem-solving ability. Our findings indicate that TBL interventions were associated with statistically significant improvements in all four competencies, with the largest effect observed in problem-solving ability (SMD = 1.10, CI = 0.37–1.83, *p* = 0.006). While the effect on self-efficacy was the smallest (SMD = 0.32, CI = 0.13–0.51, *p* = 0.005), it remained significant. Additionally, a high degree of heterogeneity was observed across studies, necessitating further investigation into the moderating factors influencing TBL effectiveness.

### 4.2. Comparison with Previous Studies

[1] ([1]) conducted a systematic review to determine the impact of TBL by incorporating learning outcomes as the primary variable and including both Italian and English literature. They included papers from many more countries compared to this study, which included papers from only Korea. Among the studies included in [1] ([1]) review, only Lee’s (2018) study is identical to this study. [4] ([4]) scoping review explored the effects of TBL by reviewing 41 studies involving undergraduate students, graduate students, and clinical nurses. Our study and the five studies included in [4] ([4]) review are identical but cover a broader range of outcomes by including cohort and experimental studies. These six studies present a more diverse array of results by examining various studies on TBL in nursing education.

### 4.3. Scope and Significance of This Study

[1] ([1]) analyzed the effect of TBL interventions on only teamwork and communication, while [4] ([4]) included studies on not only undergraduate students but also graduate students and clinical nurses. These studies have limitations in confirming the effectiveness of TBL interventions for undergraduate nursing students because they included all studies in which a TBL intervention was implemented without restricting the outcome variables. In contrast, our study focused on communication ability, self-efficacy, learning ability, and problem-solving ability. These competencies are considered the most important competencies in achieving learning outcomes in the nursing education field, and nursing students are required to have these competencies. Moreover, this study conducted a meta-analysis as well. Hence, this study holds significance as the first study to be conducted and to report the results.

### 4.4. Impact of the COVID-19 Pandemic on TBL Implementation

Most previous research, including this meta-analysis, predates 2018, reflecting the state of nursing education before the onset of the coronavirus pandemic. Before the pandemic, TBL thrived in education; however, from 2020 to 2022, social distancing measures imposed significant constraints on its implementation. This study confirms that TBL was effective just before the pandemic and serves as a valuable reference point. Hence, its findings are expected to be pivotal for designing post-pandemic TBL strategies in educational settings.

### 4.5. Risk of Bias Methodological Considerations

Of the 12 studies analyzed, one employed a randomized experimental design, while the rest utilized a quasi-experimental approach. Given that most of these studies were conducted in classroom settings, there could have been limitations in subject randomization. While subject selection was unbiased in the sole randomized experimental study, there was a high risk of bias in subject allocation and performance. This was likely due to the direct involvement of the researcher in conducting TBL as an educator. To mitigate this problem, future studies should employ a double-blind methodology. Among the 11 non-randomized studies, the most significant risk of bias was in the measurement of the outcome variables. Rigorous intervention studies should be conducted to counteract the potential halo effect stemming from the researcher.

### 4.6. Variability in Learning Competency Measurements

This study examined four outcome variables, each assessed using 12 different tools, illustrating the diverse approaches employed to gauge learning competency. Learning ability was measured predominantly using the Self-Directed Learning Readiness Scale, demonstrating minimal variability. Conversely, problem-solving ability was assessed using three distinct tools, thus exhibiting the most variability. To enhance the effectiveness of future research on TBL, the tools to measure outcome variables should be selected based on prior research. This approach will not only ensure consistency in measurement tools for comparative analysis but also augment the potential for significant research outcomes by incorporating previous findings.

### 4.7. Overall Effectiveness of TBL Intervention

We conducted a meta-analysis to assess the overall learning competency of nursing students before and after the TBL intervention. A substantial effect size of 0.69 was found, indicating the effectiveness of TBL interventions in enhancing the learning competency of nursing students. This finding aligns with that of [14] ([14]), who conducted a systematic literature review comparing TBL with lecture-based education. They found TBL to be a potent teaching approach for enhancing nursing students’ academic proficiency and various competencies. [1] ([1]) examined the impact of TBL on nursing education and found it to be better than traditional lecture formats in fostering academic performance and nursing-related skill development. TBL, which is characterized by active learning, pushes students to apply conceptual knowledge through a sequence of activities that include individual tasks, teamwork, and immediate feedback ([12]). Essentially, TBL enhances not only the collaborative but also the individual competencies of nursing students. Thus, future assessments of TBL’s efficacy in nursing education should incorporate the measures of team, as well as individual competencies.

### 4.8. Impact of TBL on Communication Ability

Our analysis of TBL’s effect on specific competencies revealed a significant effect size for communication ability (SMD = 0.74). [1] ([1]) meta-analysis further reinforces the notion that TBL contributes to enhancing communication skills. Despite being a specific area of competence, communication skills are honed through interpersonal interactions, making their development iterative. Thus, it is plausible to infer that TBL, which is rooted in collaborative endeavors, fosters the refinement of communication skills. Nonetheless, a considerable degree of heterogeneity (90%) was observed in effect sizes. This variation could have stemmed from the diverse array of assessment tools employed to gauge communication skills. Additionally, the studies included in this meta-analysis utilized tools designed for the general population or college attendees rather than those specifically tailored for nursing students. To interact with patients, nursing demands a high caliber of communication skills, particularly therapeutic communication skills, and this communication entails not only verbal but also non-verbal elements ([5]). Thus, future assessments of nursing students’ communication abilities should employ tools that align with the nuanced requirements of nursing.

### 4.9. Impact of TBL

The effect on self-efficacy was found to be modest at SMD = 0.32. This effect size was the smallest among all learning competencies; nevertheless, it was statistically significant. The degree of heterogeneity was low at 16%. Meanwhile, the effect on learning ability was larger at SMD = 0.44 with no heterogeneity. Given that TBL emphasizes collaborative learning, its influence on self-efficacy and learning ability—both deeply personal competencies within the scope of this study—may appear to be comparatively subdued. Nonetheless, these effects were found to be statistically significant. TBL encompasses several stages: teacher-led preparatory study, test-based assessment of one’s mastery of core knowledge, and practical application of the acquired knowledge ([7]). Students engage in discussions and support and motivate their team members based on the knowledge acquired, ultimately arriving at solutions for the challenges presented. This process not only hones team-related skills but also improves individual self-efficacy ([10]). Considering that nursing education aims to enhance both collective and individual learning capacities, the finding that TBL effectively bolsters self-efficacy is compelling in the nursing domain. Given that this study affirms TBL’s potential in enhancing individual learning capabilities, future applications of TBL should be designed to explicitly measure improvements in both individual and team proficiencies.

The effect of TBL interventions on problem-solving ability stood out from its effect on other competencies, with an effect size of 1.2 and a notable 95% of heterogeneity. This competence featured prominently in 7 of the 12 papers and was underscored as a pivotal learning skill in TBL environments. This emphasis may stem from the fact that, as future caregivers, nursing students must hone their critical thinking and problem-solving processes. These skills are indispensable for addressing the myriad of health issues that affect patients. TBL, a pedagogical approach centered on collaborative group dynamics, has been lauded for its efficacy in cultivating critical thinking and problem-solving skills among students ([12]; [13]). Leveraging previous knowledge and peer feedback, TBL offers distinct advantages over traditional lecture-based methods, particularly when the professor-to-student ratio is disparate ([2]; [9]; [13]). It stands to reason that TBL enhances nursing students’ problem-solving abilities by guiding them through the process of drawing conclusions and devising solutions within a supportive group setting. Hence, the adoption of TBL as a teaching methodology holds promise for enhancing nursing students’ problem-solving abilities.

### 4.10. Influence of Intervention Frequency on Learning Outcomes

We examined the differences in the impact of TBL interventions on learning competencies based on the frequency of the interventions. Interventions that were provided fewer than four times were more effective in enhancing communication ability, self-efficacy, and problem-solving ability than those that were provided more than four times. This result indicates that interventions with fewer sessions, which are typically focused on one or two topics or scenarios, are more beneficial. Conversely, interventions with four or more sessions, covering multiple scenarios or topics, may lead to academic fatigue and reduced participation. Considering the demanding nature of nursing education, in which students juggle numerous classes and extensive study requirements, individualized learning and increased interaction with peers and instructors are crucial for effective participation in TBL classes. It may be assumed that academic performance declines because of personal fatigue or relationship strain, but the evidence supporting this notion is lacking. In addition to evaluating the efficacy of TBL classes, it is crucial to monitor their impact on students’ academic workload. Regarding the effect of TBL interventions on learning ability, the effect improved significantly when the intervention was administered four or more times. This could be attributed to the fact that, concerning academic prowess, knowledge and skills vary from one individual to another as the number of sessions increases. Hence, when strategizing the implementation of TBL, it is imperative to consider not only its positive outcomes but also its potential drawbacks and apply it judiciously.

### 4.11. TBL-Only vs. Multicomponent TBL Intervention

We also analyzed differences in the effect of TBL interventions on learning competencies based on the type of intervention. The results revealed that communication, learning, and problem-solving abilities are notably enhanced when a TBL-only intervention is administered compared to when a multicomponent TBL intervention (such as TBL combined with simulation or reflective methods) is administered. Interestingly, the effect on self-efficacy remained similar across both types of TBL interventions. This result underscores not only the efficacy of TBL in bolstering nursing students’ capacities but also its standalone effectiveness in honing communication, learning, and problem-solving abilities. Nevertheless, it is crucial to note that this comparison solely involved TBL versus a mixed method approach and did not juxtapose educational methods utilizing simulation practice or reflective methods with TBL. Thus, a cautious interpretation of the findings is warranted. Moreover, given that this study confirmed TBL’s effectiveness for four designated core competencies, future research should explore its impact alongside mixed interventions on other skill sets.

## 5. Limitations

This review is limited in its generalizability as all included studies were conducted in Korea. The predominance of Korean studies may be due to the increasing adoption of TBL methodologies in Korean nursing education. However, it is essential to conduct further research in diverse educational and cultural settings to determine whether these findings can be extrapolated to nursing students in other countries. We acknowledge this geographic limitation and emphasize the need for cross-cultural comparative studies to evaluate the effectiveness of TBL in various academic environments. Future studies should also consider different pedagogical adaptations of TBL in international nursing curricula.

Second, this study included a limited number of randomized controlled experimental studies and a majority of non-randomized studies with a high risk of bias. Thus, generalization should be approached with caution.

Third, one of the key limitations of this review is the considerable variation in the duration (1–6 h per session) and frequency (1–13 times per week) of TBL interventions across studies. This variability makes it difficult to determine the optimal ‘dosage’ of TBL that maximizes learning outcomes. While some studies suggest that shorter, intensive interventions enhance engagement and problem-solving ability, others indicate that extended exposure to TBL fosters deeper learning and critical thinking. Future research should conduct controlled trials comparing different durations and frequencies of TBL interventions to identify the most effective implementation strategy in nursing education.

While this review confirms the effectiveness of TBL in enhancing key learning competencies among nursing students, its findings should be interpreted with caution due to the geographic limitation of the included studies. Given that all studies were conducted in Korea, further research is needed to explore the applicability of TBL in diverse educational and cultural settings.

## 6. Conclusions

This systematic review and meta-analysis confirm that team-based learning (TBL) enhances nursing students’ communication ability, self-efficacy, learning ability, and problem-solving skills. Given that all studies were conducted in Korea, further research is needed to assess TBL’s effectiveness in diverse educational contexts. Additionally, exploring the optimal duration, frequency, and structure of TBL interventions will help refine its application. More rigorous experimental studies, including randomized controlled trials, are necessary to validate its superiority over other teaching methods. Educators should integrate TBL strategically to maximize student engagement and learning outcomes.

## Figures and Tables

**Figure 1 behavsci-15-00390-f001:**
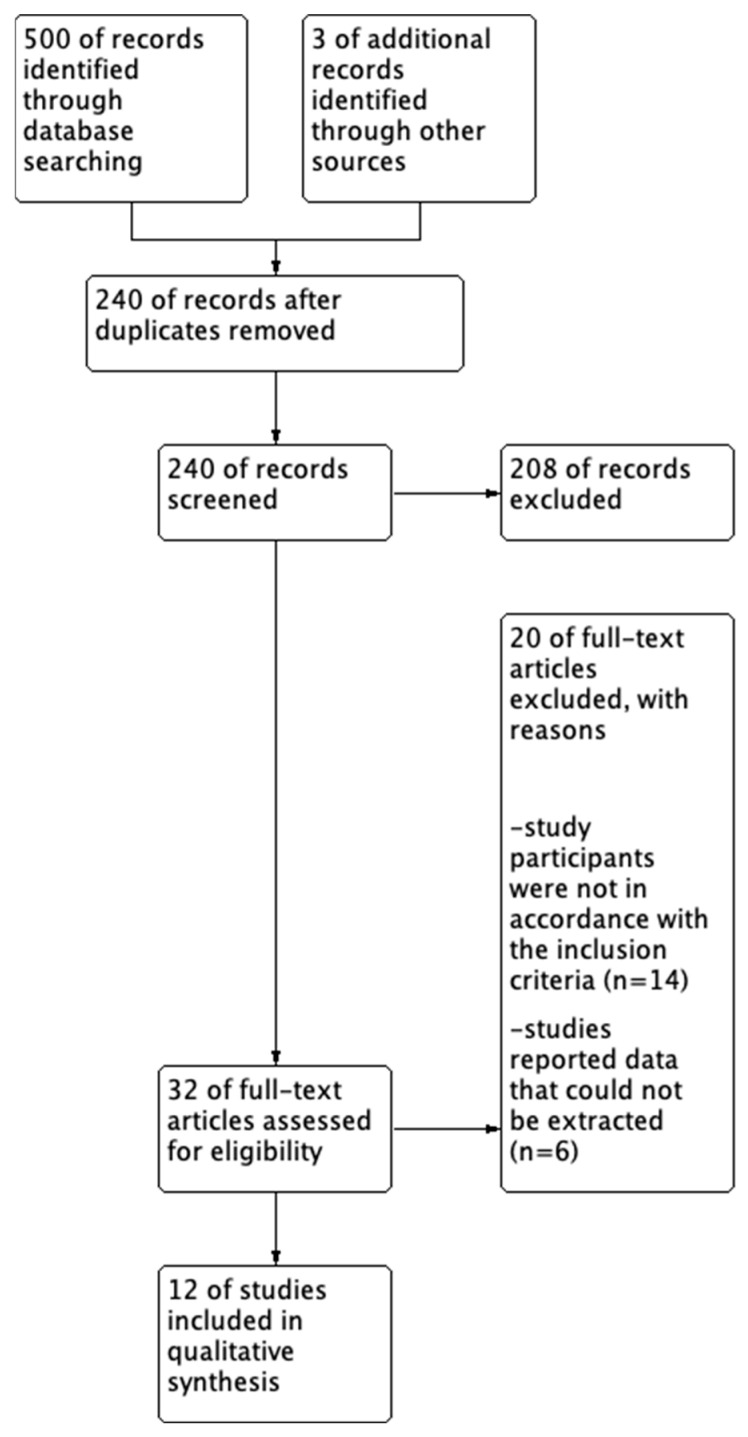
The process of the literature search based on the PRISMA statement.

**Figure 2 behavsci-15-00390-f002:**
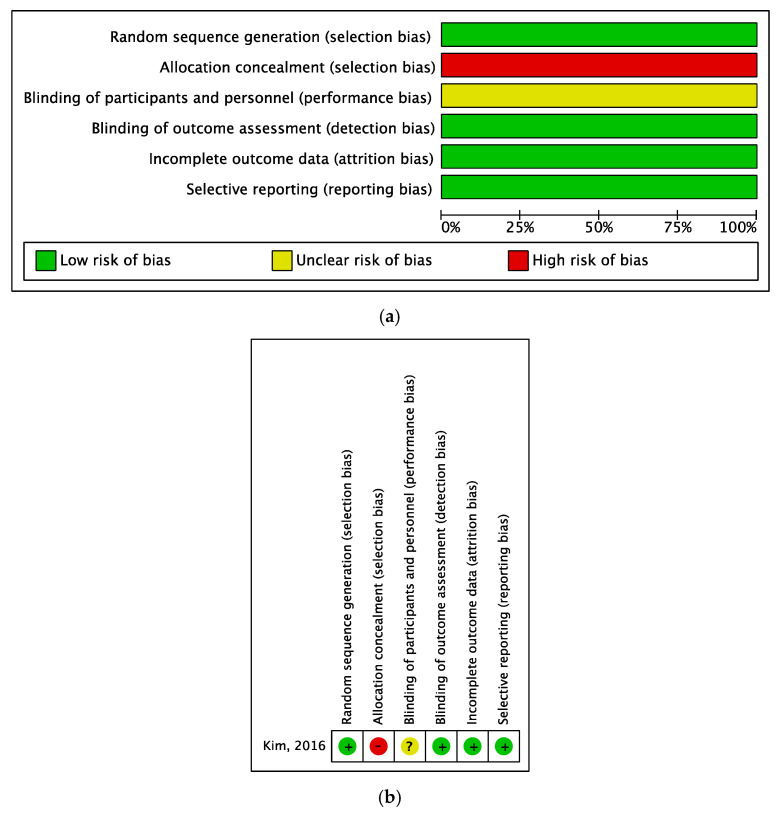
(**a**) Risk of bias summary of RCT. (**b**) Risk of bias graph of RCT.

**Figure 3 behavsci-15-00390-f003:**
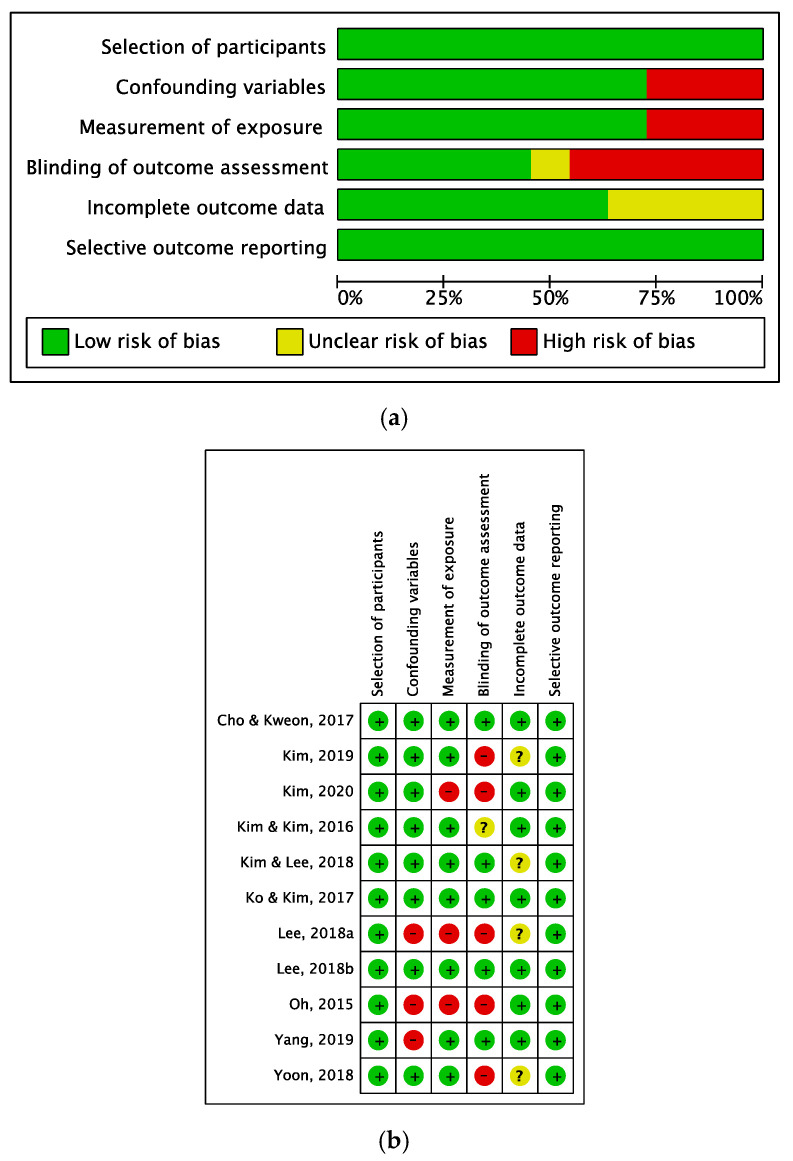
(**a**) Risk of bias summary of non-RCT. (**b**) Risk of bias graph of non-RCT.

**Figure 4 behavsci-15-00390-f004:**
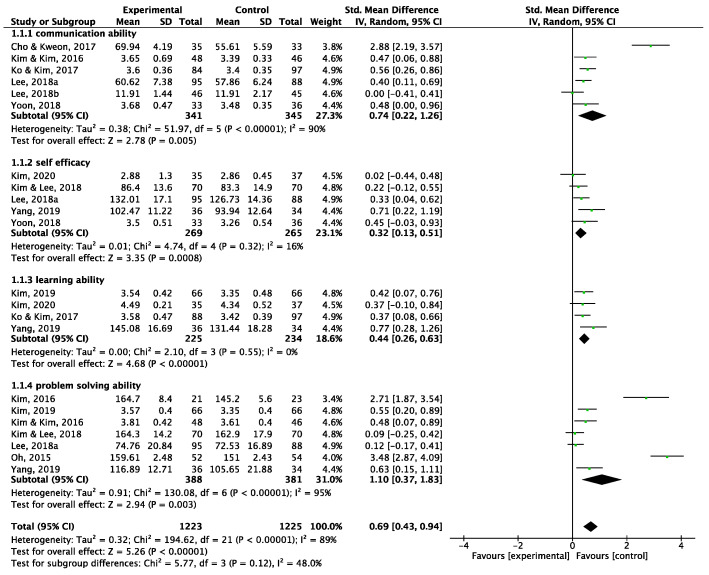
Forest plot for overall and subgroup effect size.

**Figure 5 behavsci-15-00390-f005:**
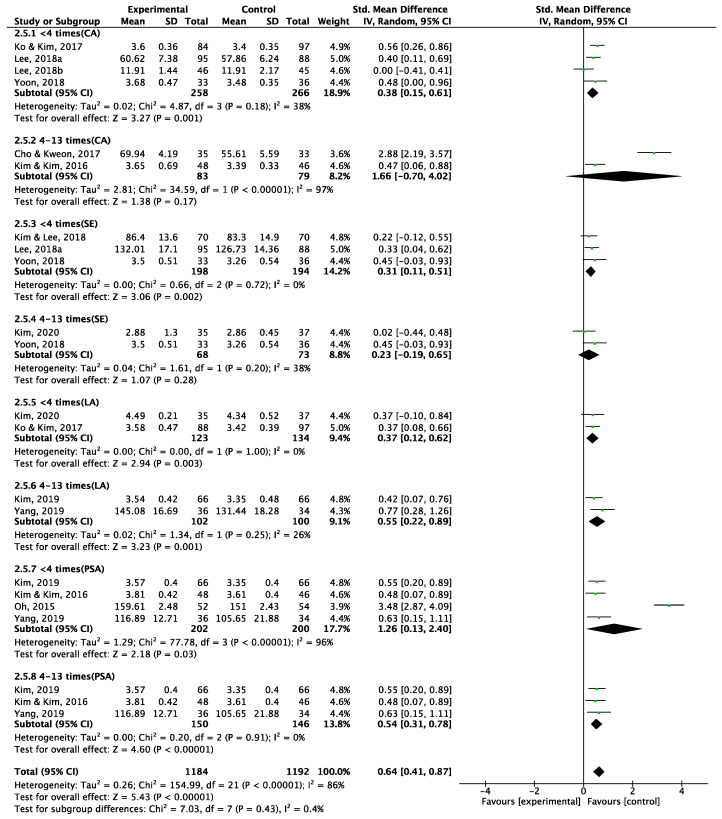
Forest plot for subgroup effect size for times of intervention.

**Figure 6 behavsci-15-00390-f006:**
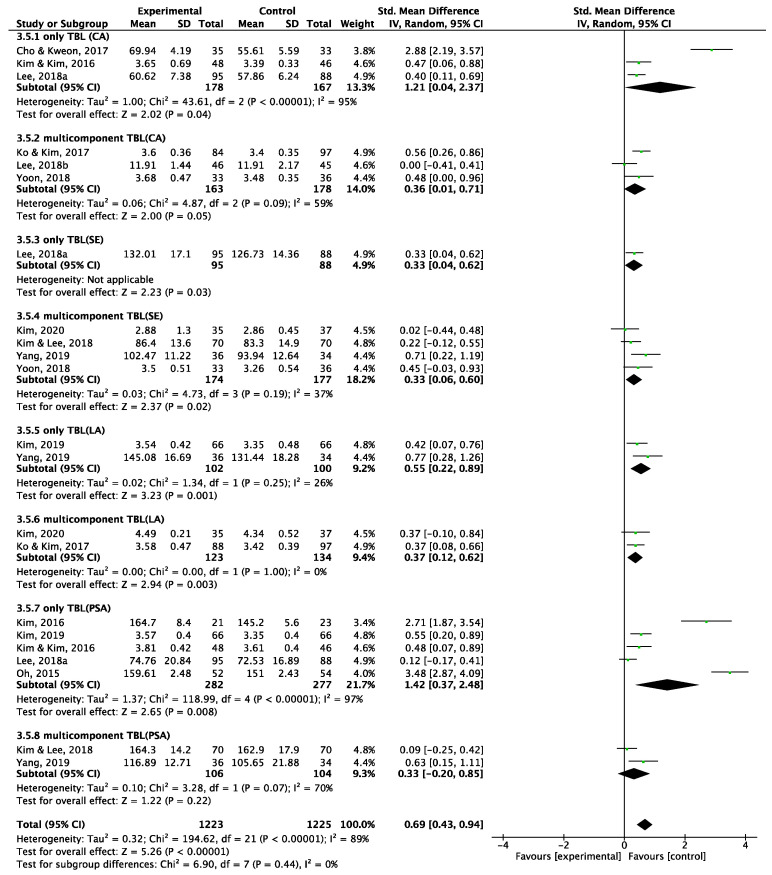
Forest plot for subgroup effect size for type of intervention.

**Table 1 behavsci-15-00390-t001:** Characteristics of included studies.

No.	Author, Year	Country	Study Design	Type of Class	Participants		Intervention Group			Control Group	Outcomes			
					Grade	Intervention (*n*) Control (*n*)	Collaborative Competence	Duration (Hours)/ Session	Frequency (Times)	Content	Communication Ability	Self-Efficacy	Learning Ability	Problem Solving Ability
1	Kim, 2019	Korea	Quasi	Adult health nursing	Junior	Inter. (66) Control (66)	TBL	2	6	Lecture and scenario-based learning			SDLRS	PSC
2	Oh, 2015	Korea	Quasi	Adult health nursing	Sophomore	Inter. (52) Control (54)	TBL	3	2	Lecture and scenario-based learning				PSC
3	Yang, 2019	Korea	Quasi—no-synchronized design	Children’s health nursing	Junior	Inter. (37) Control (37)	Team-based problem-centered learning program applying smart education	2	7	Lecture based learning		ASE	SDLRS	PSC
4	Kim & Lee, 2018	Korea	Quasi	Woman’s health nursing	Junior	Inter. (70) Control (70)	Simulation practice training combined with team-based learning	2	2	Lecture based learning		ASE		PSC
5	Kim, 2020	Korea	Quasi	Simulation practice	Senior	Inter. (35) Control (37)	Team-based simulation learning using SBAR	2	4	Simulation-based education		SES	CPS	
6	Yoon & Lee, 2018	Korea	Quasi	Fundamental nursing	Sophomore	Inter. (33) Control (36)	TBL-based simulation	6	2	Lecture-based learning	Communication clarity	SCCT		
7	Lee, 2018	Korea	Quasi	Woman’s health simulation practice	Junior	Inter. (46) Control (45)	TBL-based simulation	1	1	Simulation-based education	CPS			
8	Kim & Kim, 2016	Korea	Quasi	Fundamental nursing	Sophomore	Inter. (48) Control (46)	TBL	1	13	Lecture-based learning	GICC-15			PSC
9	Ko & Kim, 2017	Korea	Quasi	Simulation practice	Senior	Inter. (84) Control (84)	SBE combined TBL	3	3	Simulation-based education	CSS		SDLRS	
10	Cho & Kweon, 2017	Korea	Quasi	Communication	Sophomore	Inter. (37) Control (37)	TBL program	2	5	Lecture-based learning	CES GISS15			
11	Lee, 2018	Korea	English	Adult health nursing	Senior	Inter. (95) Control (88)	TBL program	2	3	Lecture-based learning	GISS15	SL		PSC CPS
12	Kim, 2016	Korea	RCT	Adult health nursing	Junior	Inter. (32) Control (31)	TBL	2	3	Lecture-based learning				PSC

Quasi: quasi-experimental nonequivalent control group pretest–posttest design; RCT: randomized controlled trial; SDLRS: Self-Directed Learning Readiness Scale; PSC: problem solving scale for college students; SES: Self-Efficacy Scale; ASE: Academic self-efficacy; CPS: clinical performance scale; GISS15: global interpersonal communication competence scale; CES: Communication efficacy Scale; SL: self-leadership; SCCT: self-confidence of critical thinking; Communication skills in college students and adults Scale.

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
