# Peer review of "Effect of Team-Based Learning Interventions on the Learning Competency of Korean Nursing Students: A Systematic Review and Meta-Analysis"

_behavsci, 2025, doi:10.3390/bs15030390_

Round 1

Reviewer 1 Report (Previous Reviewer 2)

Comments and Suggestions for Authors

Dear Authors,

I am satisfied that you have addressed each of my comments. Thank you for taking the time to do this and I hope they were helpful.

Author Response

Dear Reviewer,

Thank you very much for your valuable feedback.

As per your comments, we have revised the references section to fully comply with APA 7 guidelines, and all modifications have been clearly indicated in red within the manuscript. Additionally, we have carefully reviewed the overall formatting to enhance the consistency of the document.

Your detailed review and constructive suggestions have greatly contributed to improving our manuscript.
Once again, we sincerely appreciate your time and effort.

Best regards,
Seyeon Park

Reviewer 2 Report (New Reviewer)

Comments and Suggestions for Authors

In regard to clarity, relevance, and structure, the manuscript is clear and presents a structured argument. The research topic (team-based learning (TBL) in Korean nursing education) is relevant to both nursing and educational research. The introduction effectively frames the problem and research questions, followed by a well-structured (and well detailed) methods section. The results and discussion sections follow logically, providing a detailed interpretation of the findings. The references are mostly recent (within the last five years), with several citations from 2021–2023. The manuscript does not appear to have excessive self-citations. Some earlier references (e.g., 2010, 2012) are cited, but they seem foundational to the topic.

The manuscript follows a systematic review and meta-analysis methodology, using PRISMA guidelines. The search strategy and inclusion criteria are clearly described. The experimental design (randomized controlled trials and quasi-experimental studies) is appropriate for testing the hypothesis. Risk of bias assessment is well-documented, but many included studies were non-randomized, leading to potential bias in subject allocation. As mentioned above, the methods section provides detailed descriptions of databases, inclusion/exclusion criteria, and statistical analyses. The meta-analysis uses RevMan software, standardized mean differences (SMDs), and a random-effects model (REM), which is appropriate for handling heterogeneity. The study selection and data extraction processes are clearly explained, enhancing reproducibility. The PRISMA flow diagram is useful for tracking study selection. Tables summarizing study characteristics and outcome measures are appropriate. Statistical analysis is detailed, with discussion on heterogeneity (I² values), effect sizes, and subgroup analyses. However, some results (e.g., effect of intervention frequency on self-efficacy) could be strengthened with more discussion on why fewer interventions were more effective.

The conclusions are supported by the data and arguments presented. Limitations are acknowledged, including the geographic restriction to Korea and a lack of randomized controlled trials. The authors suggest future research on TBL across different cultural and educational settings, which is appropriate. The ethics statement and conflict of interest declaration are included. Funding sources are disclosed. The review is comprehensive but focuses only on Korean nursing education. A clear knowledge gap is identified (TBL’s effectiveness in different educational settings). No other recent review appears to be identical, but I did find Alberti et al. (2021) which covers related areas.

Areas for improvements woudl be to fix the random yellow highlighting throughout the document. Also, line spacing is different throughout and the references are not in APA 7 format. For example citations with multiple authors do not have the &. Journals are not italicized and the article names are not using proper capitalization. 

Author Response

Dear Reviewer,

Thank you very much for your valuable feedback.

As per your comments, we have revised the references section to fully comply with APA 7 guidelines, and all modifications have been clearly indicated in red within the manuscript. Additionally, we have carefully reviewed the overall formatting to enhance the consistency of the document.

Your detailed review and constructive suggestions have greatly contributed to improving our manuscript.
Once again, we sincerely appreciate your time and effort.

Best regards,
Seyeon Park

This manuscript is a resubmission of an earlier submission. The following is a list of the peer review reports and author responses from that submission.

Round 1

Reviewer 1 Report

Comments and Suggestions for Authors

The study conducts systematic review and meta-analysis about the Effect of team-based learning interventions on the learning competency of nursing students. The systematic review is adapted to PRIMAS standards although there are several methodological flaws that make the results biased and therefore I do not recommend its publication in Behavioral Sciences.

The information search process is not very deep and biased. The search period is very short (3 days). Generally, the search in a systematic review should be longer so that as studies are incorporated, the search can be refined with new terms that can recover studies not found in the first search. It is also advisable to search the references of the different articles retrieved to identify important studies that have not been retrieved with the terms used in the search. Finally, the grey literature databases should be checked to avoid possible publication biases.

The authors have not defined a protocol prior to conducting the systematic review. The development of a protocol is highly recommended and in fact there are databases where the protocols are registered so that they can be consulted, such as PROSPERO.

The authors define an exclusion criterion in relation to studies that are not published in English but then retrieve 8 articles in Korean. In addition, language as an exclusion criterion may introduce an important bias as described in meta-analytic literature.

The number of final studies shown in the flow chart does not correspond to the number of studies analyzed in the results (see Table 1).

It is strange that all the studies retrieved are of Korean origin when at no point in the systematic review is there any reference to the fact that the review is confined to Korean nursing students. The objective of study is defined as “In this study, we aimed to investigate the effect of TBL on individual as well as team variables among nursing students. More specifically, first, we systematically reviewed intervention studies that implemented TBL teaching methods based on the Participants, Intervention, Comparison, and Outcome (PICO) model. Second, we conducted a meta-analysis to ensure the quantitative adequacy of the studies reviewed. Finally, we confirmed heterogeneity and performed subgroup analyses” and in the model PICO: “Population: Nursing undergraduate students”. It also does not appear in the title.

Regarding statistical methodology, the authors do not justify why they choose a random effects model. Nor do they indicate which estimator of the standardized mean difference they have used: the unbiased or biased estimator. The authors indicate that they have only included studies that presented means, standard deviations and sample sizes without considering that the standardized mean difference can be obtained if the studies provide the test value or the p-value and the sample sizes.

The overall SMD value is not very reliable nor is the heterogeneity because there is a problem of outcome dependence. There are several studies that provide various effect sizes calculated from the same individuals. Therefore, the starting assumption of the statistical analyses used is that the observations are independent. For example, this affects the Q test used.

  All these factors influence the results and therefore, the conclusions of the systematic review should be viewed with caution.

Reviewer 2 Report

Comments and Suggestions for Authors

Title

Clear and appropriately indicates the focus of the manuscript and methods.

Abstract

Could the first two sentences (lines 9-11) be merged into one to increase readability? There is a clear overview of the findings – that said, I don’t think ‘all the findings were statistically significant’ is needed given the p values reported.

Introduction

In lines 51-58 the authors should elaborate further on TBL. It is mentioned that TBL ‘comprises three key phases’ yet only the ‘student preparation phase’ is explicitly outlined. Please add further detail here.

In lines 60-61 the authors also state ‘Many studies have examined the effects of TBL on nursing education, particularly on team dynamics and personal skills such as problem-solving and critical thinking’. Could the authors please elaborate here – what picture does previous literature generally paint with respect to the effects of TBL on team dynamics and personal skills?

Line 63 – technically a systematic review and meta-analysis is not a ‘study’. Please revise to ‘In this review’. Additionally, it is presumed that your systematic review would have had a question that you aimed to address. Could this be stated here?

Methods

Lines 84-85 – is there any justification for limiting the search to the most recent decade?

In the inclusion criteria the authors state ‘dissertations’ are included – does this not pose a potential issue with quality, especially given the lack of peer review?

Results

Given all of the papers are included from one country (Korea)  - I have concerns as to how well the findings can apply to other countries given differences in typical TBL approaches utilised. Should the paper be framed as a review of TBL approaches in Korea specifically?

The way the figures have been inserted are quite difficult and cumbersome to read (especially Figure 4).

Discussion

This section should begin by re-stating the aim of the review and then summarising the key findings as opposed to jumping straight into discussing prior research (see lines 303-313).

With respect to limitations, given studies varied considerably in terms of duration (1-6 hours), frequency (1-13 times per week), how can you discern the optimal TBL approach? Perhaps there needs to be more research to determine the optimal ‘dosage’ of TBL intervention.

Please avoid use of the term ‘groundbreaking’ – given all papers are from one country, and as noted there may be limited wider applicability, the use of this term does not seem justified.

Conclusion

This reads as if it has been AI generated (use of the term underscores) and is quite a generic conclusion. This should be re-written, providing a clear take-home message to the reader outlining the importance of the findings and their implications.